

# Why did the storm ex-Gaston (2010) fail to redevelop during the PREDICT experiment?

T. M. Freismuth[1], B. Rutherford[2], M. A. Boothe[1], and M. T. Montgomery[1]

[1]Naval Postgraduate School, Monterey, CA, USA
[2]Northwest Research Associates, Redmond, WA, USA

Received: 3 September 2015 – Accepted: 3 December 2015 – Published: 19 January 2016

Correspondence to: M. T. Montgomery (mtmontgo@nps.edu)

Published by Copernicus Publications on behalf of the European Geosciences Union.

**ACPD**

doi:10.5194/acp-2015-692

**Why did the storm ex-Gaston (2010) fail to redevelop during the PREDICT experiment?**

T. M. Freismuth et al.

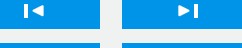
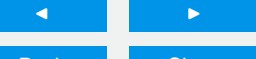

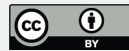

**Abstract**

An analysis is presented of the failed re-development of ex-Gaston during the 2010 PREDICT field campaign based on the European Centre for Medium Range Weather Forecast (ECMWF) analyses. We analyze the dynamics and kinematics of ex-Gaston to investigate the role of dry, environmental air in the failed redevelopment. The flow topology defined by the calculation of particle trajectories shows that ex-Gaston's pouch was vulnerable to dry, environmental air on all days of observations. As early as 12:00 UTC 2 September 2010, a dry layer at and above 600 hPa results in a decrease in the vertical mass flux and vertical, relative vorticity. These findings support the hypothesis that entrained, dry air near 600 hPa thwarted convective updraughts and vertical mass flux, which in turn led to a reduction in vorticity and a compromised pouch at these middle levels. A compromised pouch allows further intrusion of dry air and inhibits vorticity amplification. This study supports recent work investigating the role of dry air in moist convection during tropical cyclogenesis.

## 1   Introduction

Recent work has established a new overarching framework for understanding tropical cyclone formation from easterly waves (Dunkerton et al., 2008, hereafter DMW09). This framework, for describing how such hybrid wave-vortex structures develop into tropical depressions, was likened to the development of a marsupial infant in its mother's pouch. By analogy, a juvenile proto-vortex is carried along by its parent wave until the proto-vortex is strengthened into a self-sustaining entity. For tropical storms developing from within tropical waves, the recirculating flow in the wave's critical layer corresponds to the "wave-pouch". Here, the wave and mean-flow speeds are similar, along a critical latitude oriented approximately parallel to the easterly jet, and the trough axis intersects

Discussion Paper | Discussion Paper | Discussion Paper | Discussion Paper

**ACPD**

doi:10.5194/acp-2015-692

**Why did the storm ex-Gaston (2010) fail to redevelop during the PREDICT experiment?**

T. M. Freismuth et al.

meridionally. Storm formation occurs typically near the intersection of critical latitude and trough axis.[1]

The new cyclogenesis model and accompanying scientific hypotheses were established observationally in the Atlantic and eastern Pacific sectors by DMW09. They find additional support in idealized numerical modeling simulations (Wang et al., 2010a, b; Montgomery et al., 2010b; Nicholls and Montgomery, 2013), recent case studies in the field in the western North Pacific during the Tropical Cyclone Structure Experiment 2008 (TCS08, Montgomery et al., 2010a; Lussier III, 2010; Montgomery et al., 2012; Raymond and Lopez-Carrillo, 2011; Lussier III et al., 2014), in the Atlantic during the Pre-Depression Investigation of Cloud Systems in the Tropics (PREDICT) campaign in 2010 (Montgomery et al., 2012; Smith and Montgomery, 2012; Davis and Ahijevych, 2012, 2013), in NASA's ongoing Hurricane and Severe Storm (HS3) missions (2012–2015) and the case of Hurricane Sandy (Lussier III et al., 2015). The field data afford a resolved view of horizontal and vertical structure in the wave pouch and its immediate surroundings, valuable for system centering, circulation magnitude, vorticity balance, interleaving of air masses, and moist thermodynamic profiles.

A corollary from the new model is that the non-development of a candidate tropical disturbance is linked to the pouch structure being compromised. Currently, it is thought there are two principal ways the pouch can be compromised. The first way is a kinematic effect caused by the differential shearing of the pouch in the vertical plane or horizontal plane, or both. Shearing of the pouch degrades the protective womb that both nurtures the incipient proto-vortex and supports deep convective activity. The second way is a combined thermodynamic-dynamic effect associated with the intrusion of dry air (so-called "anti-fuel") into the otherwise moist pouch from a relatively dry environment. The injection of anti-fuel into the low-to-mid tropospheric wave-pouch acts to limit the vigor of deep convection and the amplification of vertical vorticity in convec-

---

[1]The jet contains two such critical latitudes, the cyclonic one equatorward of the jet axis being instrumental to storm formation, the anticyclonic one poleward of the jet axis relevant to dusty Saharan air outbreaks and dry subsidence aloft.

**ACPD**

doi:10.5194/acp-2015-692

**Why did the storm ex-Gaston (2010) fail to redevelop during the PREDICT experiment?**

T. M. Freismuth et al.

Title Page

| Abstract | Introduction |
| Conclusions | References |
| Tables | Figures |

|◄ | ►|
◄ | ►
Back | Close

Full Screen / Esc

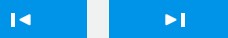

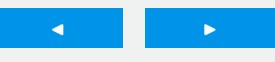

Interactive Discussion

tive updraughts (Kilroy and Smith, 2012), which is essential for spinning up a tropical cyclone (Smith and Montgomery, 2012).

The non-developing case of ex-Gaston (2010) during the PREDICT experiment is arguably one of the most extensively observed non-developing tropical disturbances ever. The five consecutive days of observational data for such a non-developing disturbance is unprecedented.

Based on the foregoing discussion, there remains an important question in understanding the non-development of ex-Gaston: Did ex-Gaston have a robust (closed), protective pouch? If ex-Gaston did, in fact, have a robust pouch, one would expect the system to redevelop and possibly intensify. We will show that ambient vertical shear and the entrainment of dry, environmental air early on 2 September led to the degradation of ex-Gaston's pouch and this plagued the convection within the pouch for the entire observational period of the PREDICT experiment.

## 2  Review of Pre-PREDICT Gaston

Tropical Storm Gaston developed from an African easterly wave that moved westward from the African coast on 28 August 2010. The National Hurricane Center (NHC) designated Gaston as a tropical storm at 15:00 UTC 1 September. Despite being in a favorable environment with relatively low vertical shear (discussed further below) and a SST of 28.5 °C (Gjorgjievska and Raymond, 2014), convection associated with Gaston quickly diminished, and NHC downgraded the system to a post-tropical/remnant low by 21:00 UTC 2 September. Convective activity increased on 3 September, however it did not re-organize and the system remained a remnant low.

## 3  Data sources

This study uses the European Centre for Medium-Range Weather Forecasts (ECMWF) analyses from 28 August 2010 to 11 September 2010. The analysis fields have a hor-

Discussion Paper | Discussion Paper | Discussion Paper | Discussion Paper

**[ACPD](doi:10.5194/acp-2015-692)**

doi:10.5194/acp-2015-692

**Why did the storm ex-Gaston (2010) fail to redevelop during the PREDICT experiment?**

T. M. Freismuth et al.

izontal resolution of 0.25°, 25 vertical levels from 1 to 1000 hPa, and temporal output every 6 h. Dropsonde data from the Pre-Depression Investigation of Cloud-Systems in the Tropics (PREDICT) Experiment were included in the standard assimilation system at ECMWF.

The PREDICT Experiment, as described in Montgomery et al. (2012), was a dedicated field study that set out to acquire empirical data to quantify thermodynamic and kinematic parameters in developing and non-developing tropical disturbances in the Atlantic Ocean. The primary platform for this experiment was the NSF-NCAR Gulfstream V (GV) with EOL/Vaisala GPS dropsondes. The GV was able to make drops
from altitudes as high as ∼ 13 km. There were 5 research flights with 109 dropsondes conducted during ex-Gaston (Fig. 1).

## 4   Results

We begin our analysis by characterizing the vertical shear that affected Gaston's pouch. "Deep-layer shear" and "pouch shear" are computed by taking the vector differential
of horizontal winds between the 200 and 850 hPa levels, and between the 500 and 850 hPa levels, respectively, averaged over a a $3° \times 3°$ box centered at the pouch center. The pouch-scale averaging is performed on a $3° \times 3°$ box, centered on the circulation center as defined by the 700 hPa tracking level.

For both the deep and pouch shear, the magnitude of the shear decreases rapidly
from ∼ 20 m s$^{-1}$ on 30 August to ∼ 2 m s$^{-1}$ on 2 September (Fig. 2). During the same period, the direction of the deep and pouch shear shifts from easterly to westerly flow (Fig. 3). After 2 September, the magnitude of the shear (deep and pouch) increases to ∼ 5 m s$^{-1}$. The pouch shear direction slowly becomes more northerly by 5 September. The deep shear, though, rapidly changes direction from northeasterly to southwest-
erly from 12:00 UTC 2 September to 00:00 UTC 3 September, in the ECMWF data. The deep shear returns to an easterly flow on 4 September. These shear results are consistent with the analysis of PREDICT data by Davis and Ahijevych (2012). The

**ACPD**

doi:10.5194/acp-2015-692

**Why did the storm ex-Gaston (2010) fail to redevelop during the PREDICT experiment?**

T. M. Freismuth et al.

National Hurricane Center defines vertical shear of $12\,\mathrm{m\,s}^{-1}$ as an upper limit for favorable conditions for tropical cyclogenesis. The magnitude of the vertical shear (typically $4$–$8\,\mathrm{m\,s}^{-1}$) for ex-Gaston, while below this heuristic limit for a SST of $28.5\,^\circ$C, does suggest lateral flow and a potential contribution of dipole-like distribution of vorticity
from a non-advective flux (Haynes and McIntyre, 1987; Raymond et al., 2014). This contribution could be a net increase or decrease of vorticity.

The evolution of other pertinent variables is shown in Fig. 4. In the subpanels of this figure we show a time-height Hovmoeller diagram of relative humidity, relative vorticity ($\zeta$), and vertical mass flux at each level from averages taken over a $3^\circ \times 3^\circ$ box (referred
to as pouch scales). Similar analysis was done for a $1^\circ \times 1^\circ$ box (referred to as sub-pouch scale) centered on the circulation center; trends were similar to those for the $3^\circ$ box, but are not shown. We use the model vertical velocity in pressure coordinates, $\omega$, to calculate the mass flux as $\rho w = -\omega/g$, where $\rho$ is density, $w$ is vertical velocity in height coordinates, and $g$ is the acceleration due to gravity. On both scales in the
ECMWF data, a layer of dry air above $600\,\mathrm{hPa}$ appears to penetrate the pouch region on 2 September, and that dry layer persists through the decline of the system (Fig. 4). Coincident with the intrusion of the dry air are system-scale decreases in relativity vorticity and mass flux.

To gain insight into the apparent intrusion of dry air into the pouch beginning near
06:00 UTC 2 September (discussed above), we first examine the flow topology of ex-Gaston using the dividing streamline methodology discussed in Riemer and Montgomery (2011). This methodology assumes for simplicity that the flow is steady in a translating frame. Although the observed flow will be shown later to have an important transient component, this technique can provide a first look into the existing flow
topology around ex-Gaston's pouch. Figure 5 shows the horizontal flow fields and calculated dividing streamlines at 18:00 UTC 2 September from ECMWF analysis data at 500, 700, 850, and 925 hPa levels when dry air was greatly impacting the pouch.

At 700, 850, and 925 hPa, a hyperbolic point lies east of the circulation center. However, to the west of the circulation center, the pouch is open to the environment; thereby

Discussion Paper | Discussion Paper | Discussion Paper | Discussion Paper | Discussion Paper |

**ACPD**

doi:10.5194/acp-2015-692

**Why did the storm ex-Gaston (2010) fail to redevelop during the PREDICT experiment?**

T. M. Freismuth et al.

providing a pathway for air parcels to enter the pouch (i.e., an "open pouch"). At 500 hPa the hyperbolic point is northwest of the circulation center, and the pouch is open to the east. This interpretation is consistent with the study of RM12 who performed a more comprehensive study of the flow topology of ex-Gaston. In particular, RM12 analyzed

Lagrangian coherent structures derived from particle trajectories, and found that lateral, dry air intrusion occurred from 1 to 5 September (RM12 Fig. 6 therein). These current findings at 700, 850, and 925 hPa are consistent with the detailed Lagrangian analysis of ex-Gaston by RM12.

We can further study the structure of the pouch by looking at the time-dependent

nature of the flow by calculating hyperbolic trajectories (Samelson and Wiggins, 2006). Hyperbolic trajectories are trajectories of the time-independent flow field that share the same linear stability properties as hyperbolic fixed points in time-independent flow. These hyperbolic trajectories have stable and unstable manifolds associated with them, and these manifolds control particle transport in time-dependent flow (Ide et al., 2002).

Figure 6 shows a time sequence of stable and unstable manifolds at 500 hPa (left column) and 700 hPa (right column) from 00:00 UTC 1 September to 00:00 UTC 3 September. Stable manifolds are indicated with red lines, and unstable manifolds are indicated with blue and cyan lines. For reference, a 3° radius circle around ex-Gaston's diagnosed pouch center is indicated by the green circle. Throughout this 48 h period,

the stable manifold (red line) and an unstable manifold (blue line) intersect east of ex-Gaston's pouch on the 500 and 700 hPa pressure surfaces. The intersection of these manifolds marks the location of a hyperbolic trajectory, and the persistence of these manifolds is indicative of the pouch having a barrier to intrusions from the northeast, east and southeast. In this case the manifolds comprise only part of a cat's eye. At

700 hPa, the stable (red line) manifold also intersects the unstable (cyan line) manifold south of ex-Gaston's pouch. We do not observe this second intersection on the 500 hPa pressure level. At both the 500 and 700 hPa levels, there are no intersecting manifolds west of the pouch. No intersection implies no additional hyperbolic trajectory, and

**ACPD**

doi:10.5194/acp-2015-692

**Why did the storm ex-Gaston (2010) fail to redevelop during the PREDICT experiment?**

T. M. Freismuth et al.

leaves no way for boundaries to be topologically connected into a separatrix.[2] While there is difference between the dividing streamline and Langrangian manifold analyses at 500 hPa, the two methods are consistent at the other pressure levels presented. The Langrangian manifold method is more complete and accurate by incorporating the time-dependent nature of the analyzed flow.

To identify the source region for the dry air that entered ex-Gaston's pouch at 700, 600, 500, and 400 hPa on 18:00 UTC 2 September, backward trajectories were computed for particles seeded within a 3° radius of the pouch center. Trajectories were computed as in RM12 using a fourth-order Runge–Kutta method with a 15 min intermediate time step and bi-cubic interpolation in both time and space on constant pressure

surfaces. At 400 hPa, particles that are within a 3° radius of the pouch at 18:00 UTC 2 September originated to the north of the pouch (Fig. 7). At the 500 hPa level, particles that are within a 3° radius of the pouch at 18:00 UTC 2 September originated primarily northeast and southwest of the pouch (Fig. 8).

For the trajectories identified in the foregoing figures, it is of interest to document the evolution of pseudo-equivalent potential temperature, $\theta_e$. For a moist air parcel, $\theta_e$ is approximately materially conserved in the absence of mixing processes. On a given pressure surface, $\theta_e$ is a function of moisture and temperature and because of its tracer-like property and weak temperature gradient in the tropics, increases or de-

creases in $\theta_e$ along a constant pressure trajectory reflect primarily changes in moisture. For all calculations presented here, we use the $\theta_e$ definition as given by Bolton (1980) (his Eq. 43).

Figure 9 summarizes the evolution of $\theta_e$ for the trajectories identified previously in Fig. 8 on the 500 hPa level. A colored point in the figure represents a snapshot of

the particular particle's $\theta_e$ and radial distance from the center of Gaston's pouch. The colors range from brown to blue, with brown denoting the earliest time of 00:00 UTC 31 August and blue denoting the latest time of 18:00 UTC 2 September. The quasi-

---

[2] A separatrix is a flow partitioning boundary formed by connected segments of manifolds or material curves.

**ACPD**

doi:10.5194/acp-2015-692

**Why did the storm ex-Gaston (2010) fail to redevelop during the PREDICT experiment?**

T. M. Freismuth et al.

**[ACPD](doi:10.5194/acp-2015-692)**

doi:10.5194/acp-2015-692

**Why did the storm ex-Gaston (2010) fail to redevelop during the PREDICT experiment?**

T. M. Freismuth et al.

regular pattern of blue dots between 0 and 3° radius is a manifestation of the initial seeding method for the backward trajectory calculation.

Figure 9 shows that particles seeded within the nominal pouch radius of 3° originate from two distinct source regions (brown points) outside of the pouch. The two source regions are indicated by the red dots in Fig. 8; one source region is located in an arch-like filament northeast of the pouch in a dry (low $\theta_e \sim 328\,\mathrm{K}$) environment; the other is located west and southwest of the pouch in a relatively moist environment ($\theta_e \sim 339\,\mathrm{K}$). As these particles enter ex-Gaston's pouch from 00:00 UTC 31 August to 18:00 UTC 2 September, the moist particles remain relatively moist, and the dry particles remain relatively dry. The black slanted line in Fig. 9 approximately differentiates these moist and dry trajectory paths, and its shallow slope indicates that dry air was not significantly moistened before entering the pouch. A similar analysis was performed for the 400 hPa level (not shown), and showed similar trends as the 500 hPa level. These results demonstrate that dry air was entering Gaston's pouch during this 66 h period from 00:00 UTC 31 August to 18:00 UTC 2 September.

## 5   Implications of dry air and a degraded pouch

The findings from the previous section showing dry air entering ex-Gaston's pouch motivate an important question in its non-redevelopment: What was the role of dry air in the non-development? To understand the role of the dry air entering the pouch it is useful to review previous studies of the role of dry air on convection. A new hypothesis on the role of dry air in tropical cyclogenesis was inspired in part by the work of Smith and Montgomery (2012). The authors studied the convective environments of the tropical disturbances during the PREDICT experiment. They found that a prominent difference between developing and non-developing disturbances was the difference in $\theta_e$ between the surface and 3 km. Smith and Montgomery (2012) hypothesizd that entrained, dry air weakens the convective updraughts and thereby weakens the vortex-tube stretching of ambient and local cyclonic vorticity. Weakening of the convective updraughts implies

a frustrated vorticity amplification process. The hypothesis of Smith and Montgomery (2012) stands in contrast to the traditional notion that dry air increases the strength of convective downdraughts and increases the low-level divergence that accompanies these downdraughts.

In another study of convective environments, James and Markowski (2010) investigated the role of dry air aloft on deep convection. In their numerical study, they found that in the low CAPE environments ($1500 \, \mathrm{J \, kg^{-1}}$) with a dry air layer of RH = 70 % near 700 hPa, that the updraught mass flux was reduced throughout the depth of the troposphere, and the downdraught mass flux was either unchanged or reduced.

In their numerical study of rotating convection during tropical cyclogenesis, Kilroy and Smith (2012) (hereafter referred to as KS12) investigated the role of the dry air. KS12 created an idealized sounding based on the ex-Gaston environment, and proceeded to modify the idealized sounding by injecting dry air into the mid-levels. They found through a series of experiments (summarized in their Table 2), that dry air aloft reduced the convective updraught strength and the vertical extent of the convective updraught.

KS12 also used a "moist" and a "dry" profile from the PREDICT Experiment. The moist profile was from 18:20 UTC 5 September, and had a total precipitable water (TPW) of $65.2 \, \mathrm{kg \, m^{-2}}$. The dry profile was from 14:48 UTC 5 September, and TPW = $43.5 \, \mathrm{kg \, m^{-2}}$. In the moist environment, KS12 found maximum convective updraught and downdraught velocities of 34 and $10.9 \, \mathrm{m \, s^{-1}}$, respectively, and vertical extent above 10 km. However, in the dry environment the maximum updraught velocity and downdraught velocities were 11.4 and $6.3 \, \mathrm{m \, s^{-1}}$, respectively, and the vertical extent was only $\sim 7 \, \mathrm{km}$ (see KS12 Fig. 7). Dry air reduced both the updraught and the maximum vertical extent, while the downdraught velocities were only moderately reduced, consistent with findings from the experiments with the idealized soundings. These results showed that dry air reduces cloud buoyancy, thus making mass flux profiles weaker and shallower than in a moist environment, as well as making the convective updraught less effective in amplifying vertical vorticity (Smith and Montgomery, 2012).

**ACPD**

doi:10.5194/acp-2015-692

**Why did the storm ex-Gaston (2010) fail to redevelop during the PREDICT experiment?**

T. M. Freismuth et al.

We examined the dropsonde data from 2 September, and compared the profiles to the data used by KS12. We found profiles from within the pouch on 2 September with similar characteristics as the profiles used by KS12. This detailed comparison is not shown. However Table 1 shows TPW, CAPE, and CIN from dropsondes on 2 September (see Smith and Montgomery, 2012 for a thorough analysis of the PREDICT data). The 2 September data compares well with the profiles used by KS12.

In total, the foregoing results suggest that the pouch was vulnerable to the environment with dry air penetrating the pouch and disrupting the amplification of vorticity. This analysis of the kinematic, dynamic, and thermodynamic structure of ex-Gaston in the ECMWF analysis, as well as the work of RM12 and Davis and Ahijevych (2012, Fig. 9 therein), show that ex-Gaston's pouch was misaligned, and vulnerable to environmental air as early as 2 September (Fig. 4), the day of the first PREDICT research flight into this remnant low. This dry air results in divergence near the 600 hPa level, thus causing an expanding circulation loop at these levels. From Kelvin's circulation theorem, as the loop expands the absolute vertical vorticity must decrease in order to conserve absolute circulation. A reduction in vorticity will create a compromised pouch, which will allow further intrusion of dry air and inhibit vorticity amplification.

Gjorgjievska and Raymond (2014, hereafter referred to as GR14) propose a different process that leads to the failed redevelopment of ex-Gaston. GR14 (p. 3076) hypothesize that the "severe decrease of the mid-level vortex observed between the period of Gaston 1 (2 September) and Gaston 2 (3 September) was a deciding factor for Gaston's failure to re-intensify". It is important to note that GR14, RM12, and Davis and Ahijevych (2012) agree on the decay of the mid-level vortex. GR14 hypothesize that convection was suppressed by a strong trade wind inversion, and attribute the decrease in the mid-level vorticity to this strong trade inversion and corresponding structure of the vertical mass flux profile. GR14 argue that the trade wind inversion air causes the decrease in magnitude with height of the mass flux profile. However, we contend that the intrusion of dry air at and above the 600 hPa level is responsible for the decrease in the mass flux profile.

To address the hypothesized influence of the strong trade wind inversion, it proves useful to review the thermodynamic structure of the dropsonde data collected on 2 September. The PREDICT experiment released 19 dropsondes in ex-Gaston on this day. GR14 included drop numbers 2 through 14 (see Table 1) in their 3DVAR analy-
sis and area-averaging schemes. Within their $4° \times 4°$ analysis box (their Fig. 8), only one drop (drop number 2, located northwest of ex-Gaston's pouch) shows clear evidence of a temperature inversion (Fig. 10), while 11 of the profiles show evidence of a dry layer above 600 hPa. It is unclear how one particular sounding could have such a hypothesized impact on the system-scale vorticity dynamics.

Our study of GR14 suggests that these authors appeared to overlook the implications of Davis and Ahijevych (2012) findings of a vertically sheared pouch and RM12's findings of dry air mixing into ex-Gaston's pouch between 1 and 3 September (RM12's Fig. 6), a time period spanning the first day of PREDICT observations (2 September). While GR14 acknowledge the role of a transient flow component in causing a reversal
in the sign of the vorticity tendency, they do not recognize the role of this component in modulating the transport of dry air into ex-Gaston. Thus, GR14 appear also to misinterpret the results of Smith and Montgomery (2012) and RM12: GR14 (p. 3076–3077) imply that ex-Gaston's pouch was "robust" (i.e., closed) on 2 September, and therefore unlikely that "dry air might have been drawn into the core of Gaston".

Our offered hypothesis of Gaston's non-redevelopment described above stands somewhat in contrast to the alternative hypothesis by Gjorgjievska and Raymond (2014). The data shown herein supports the view that dry air penetrated the pouch before the first flight into ex-Gaston and disrupted the amplification of vorticity at those levels where dry air intruded. GR14 agree with this hypothesis of a compromised pouch
and dry air intrusion, but only after 4 September.

The data from PREDICT research flight 9 on 2 September does show evidence for an inversion outside the GR14 analysis box for ex-Gaston's pouch (Fig. 10). To investigate the influence of the dry, trade inversion air west of ex-Gaston's pouch on 2 September, we performed a forward trajectory analysis (Fig. 11). Particles were seeded on

the 850 hPa pressure level west of the sweet spot location (where the PREDICT data show a temperature inversion) location at 12:00 UTC 2 September and integrated forward to 00:00 UTC 4 September. Nearly all of the particles are located outside of a 3° radius of the pouch center by 00:00 UTC 4 September. This analysis shows that the observed dry, trade inversion air on 2 September does not enter the pouch, and has little influence on the non-development of ex-Gaston.

## 6 Conclusions

Our study of the ECMWF analysis data demonstrates that ex-Gaston did not have a robust pouch and was open to the intrusion of environmental air at the mid to upper levels on all days of the aircraft observations. Ex-Gaston's pouch was closed to dry air intrusion at low levels. Lagrangian trajectory and manifold calculations using ECMWF analyses show that dry air did indeed penetrate the pouch. These findings support the hypothesis that entrained, dry air near 600 hPa inhibited convective updraughts and vertical mass flux, which in turn leads to a reduction in vorticity and a compromised pouch at these middle levels. A compromised pouch allows further intrusion of dry air and inhibits subsequent vorticity amplification, as described in the work of Smith and Montgomery (2012). The findings presented herein support our hypothesis that ex-Gaston's degraded pouch further led to the non-redevelopment of the system by limiting the amplification of vorticity and not providing a protected environment for sufficient vorticity aggregation, consistent with the marsupial paradigm of tropical cyclogenesis described by DMW09.

*Acknowledgements.* T. M. Freismuth acknowledges OPNAV N2/N6, CNMOC, and valuable discussions with Gerard Kilory and Tim Dunkerton. B. Rutherford acknowledges the support of NSF AGS-1432983. M. T. Montgomery acknowledges the support of NSF AGS-1313948, NOAA HFIP grant N0017315WR00048, NASA grant NNG11PK021 and the US Naval Postgraduate School. ECMWF data provided by Peter Bauer and Gerald Thomsen at ECMWF and Gerard Kilroy and Roger Smith from the Ludwig Maximilian University of Munich and the

**ACPD**

doi:10.5194/acp-2015-692

**Why did the storm ex-Gaston (2010) fail to redevelop during the PREDICT experiment?**

T. M. Freismuth et al.

Deutscher Wetterdienst. The views expressed herein are those of the authors and do not represent sponsoring agencies or institutions.

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

**Table 1.** Summary of Dropsondes from PREDICT Research Flight 9 (RF09) on 2 September.

| Drop Num. | Time (UTC) | TPW ($\mathrm{kg\,m^{-2}}$) | CAPE ($\mathrm{J\,kg^{-1}}$) | CIN ($\mathrm{J\,kg^{-1}}$) |
|---|---|---|---|---|
| 1 | 15:32 | 33.0 | 478 | 149 |
| 2 | 15:44 | 48.2 | 196 | 95 |
| 3 | 15:55 | 53.6 | 24 | 142 |
| 4 | 16:05 | 61.1 | 688 | 6 |
| 5 | 16:14 | 62.8 | 706 | 0 |
| 6 | 16:24 | 57.5 | 1047 | 9 |
| 7 | 16:37 | 58.4 | 612 | 29 |
| 8 | 16:47 | 63.0 | 1707 | 0 |
| 9 | 16:54 | 65.9 | 654 | 11 |
| 10 | 17:03 | 67.1 | 1649 | 0 |
| 11 | 17:13 | 59.9 | 1566 | 0 |
| 12 | 17:23 | 57.5 | 605 | 5 |
| 13 | 17:33 | 56.7 | 2 | 158 |
| 14 | 17:45 | 55.3 | 0 | 328 |
| 15 | 17:55 | 53.8 | 114 | 110 |
| 16 | 18:08 | 51.1 | 1155 | 14 |
| 17 | 18:18 | 35.6 | 525 | 75 |
| 18 | 18:30 | 36.1 | 285 | 143 |
| 19 | 18:43 | 38.2 | 101 | 155 |

Adapted from Smith and Montgomery (2012).

**ACPD**

doi:10.5194/acp-2015-692

**Why did the storm ex-Gaston (2010) fail to redevelop during the PREDICT experiment?**

T. M. Freismuth et al.

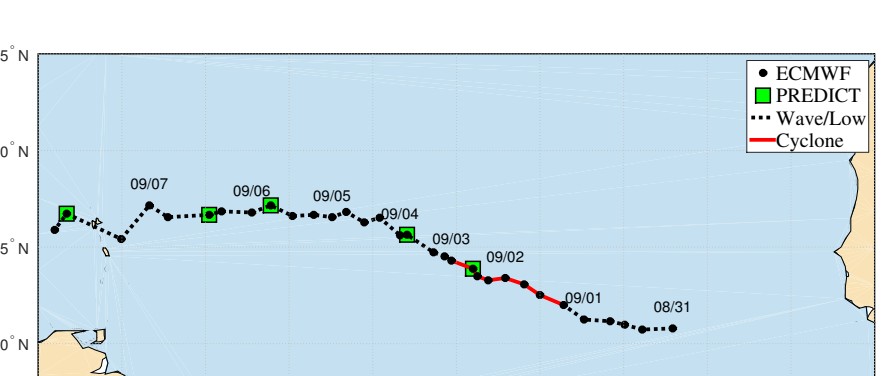

**Figure 1.** Track for ex-Gaston based on pouch center (black dots) as identified in the 6 h, ECMWF analysis data. Green squares show approximate times of PREDICT research flights over the disturbance. The red line indicates when the National Hurricane Center designated the disturbance as at least a tropical depression. The black-dashed line indicates when the disturbance was an incipient wave or remnant low.

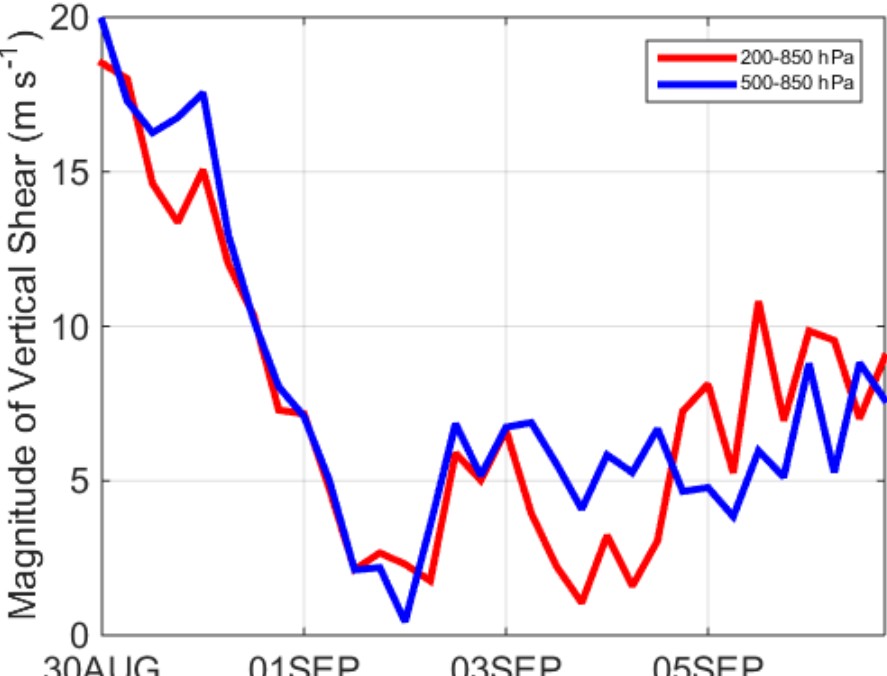

**Figure 2.** Magnitude of Vertical Wind Shear. Shear is shown for the 200–850 hPa (red line) and the 500–850 hPa (blue line) levels. Shear decreased from $\sim 20\,\mathrm{m\,s^{-1}}$ on 30 August 2010 to $\sim 2\,\mathrm{m\,s^{-1}}$ on 2 September 2010. Although the shear is below the nominal value of $12\,\mathrm{m\,s^{-1}}$, there is still persistent shear on the pouch. These results are consistent with previous studies by Davis and Ahijevych (2012).

**ACPD**

doi:10.5194/acp-2015-692

**Why did the storm ex-Gaston (2010) fail to redevelop during the PREDICT experiment?**

T. M. Freismuth et al.



Discussion Paper | Discussion Paper | Discussion Paper | Discussion Paper | Discussion Paper |

**ACPD**

doi:10.5194/acp-2015-692

**Why did the storm ex-Gaston (2010) fail to redevelop during the PREDICT experiment?**

T. M. Freismuth et al.

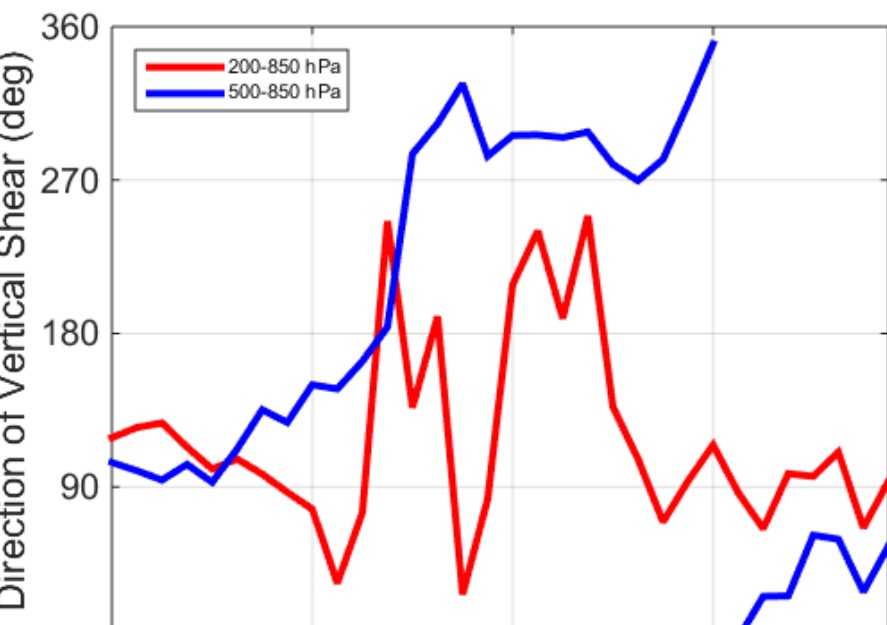

**Figure 3.** Direction of Vertical Wind Shear. The direction of the wind shear is shown for the 200–850 hPa (red line) and the 500–850 hPa (blue line) levels. The directions are compass directions in a meteorological sense. The 500–850 hPa shear is mainly from the northwest on 2 September 2010. Analysis by RM12 showed that the source region for dry air was from north of the pouch.



Discussion Paper | Discussion Paper | Discussion Paper | Discussion Paper |

**ACPD**

doi:10.5194/acp-2015-692

**Why did the storm ex-Gaston (2010) fail to redevelop during the PREDICT experiment?**

T. M. Freismuth et al.

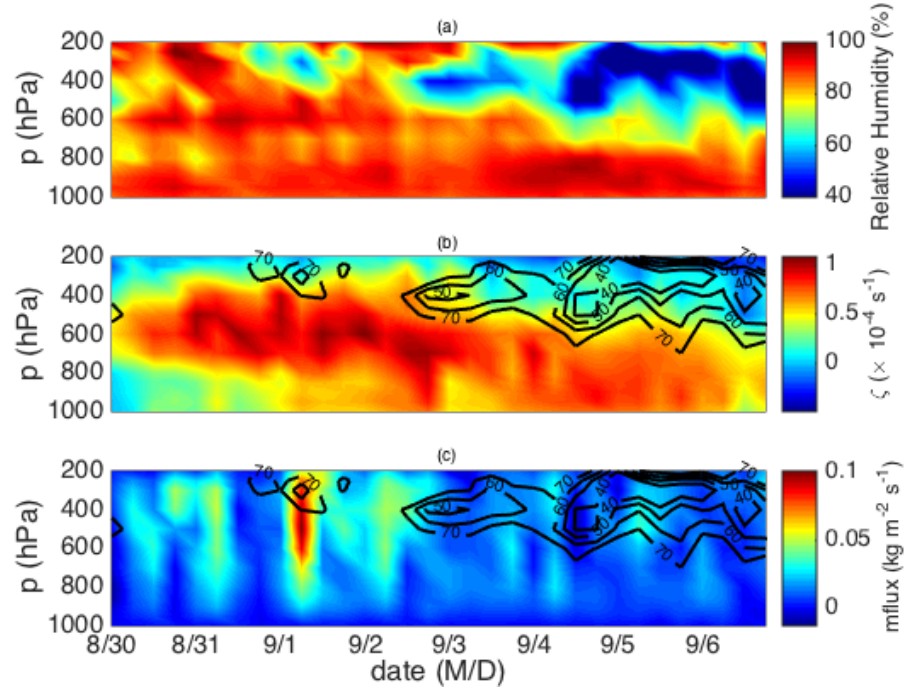

**Figure 4.** Time-height cross-section of system averaged quantities within a 3° × 3° box from 00:00 UTC 30 August to 18:00 UTC 6 September. Relativity humidity is shown in panel **(a)**, relative vorticity (ζ) with relative humidity contours of 40, 50, 60, and 70 % is shown in panel **(b)**, and mass flux with relative humidity contours of 40, 50, 60, and 70 % is shown in panel **(c)**. A dry layer near and above 600 hPa appears on 2 September and persists through 6 September. There are corresponding decreases in relative vorticity and mass flux at these times.

**ACPD**

doi:10.5194/acp-2015-692

**Why did the storm ex-Gaston (2010) fail to redevelop during the PREDICT experiment?**

T. M. Freismuth et al.

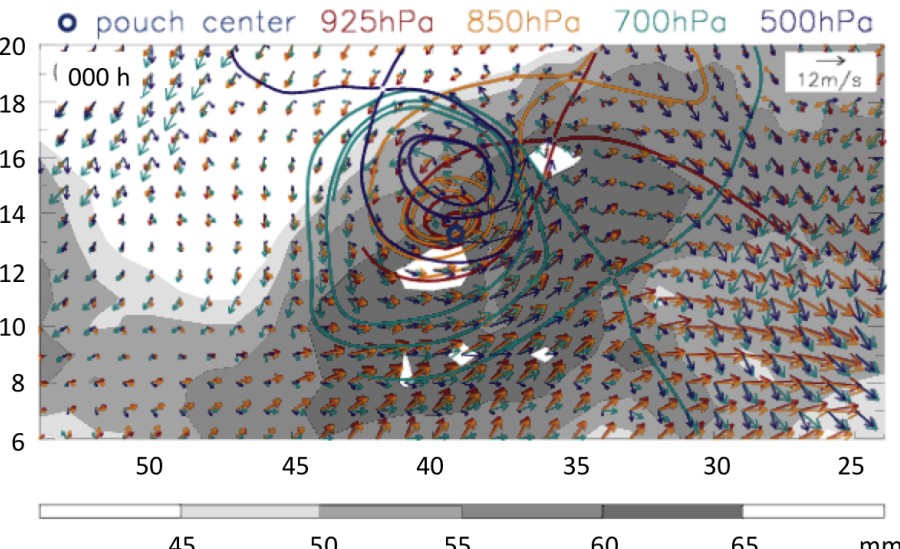

**Figure 5.** Dividing streamlines at 18:00 UTC 2 September 2010 are shown for 500, 700, 850, and 925 hPa, and are overlaid on co-moving wind vectors at each level and total precipitable water.

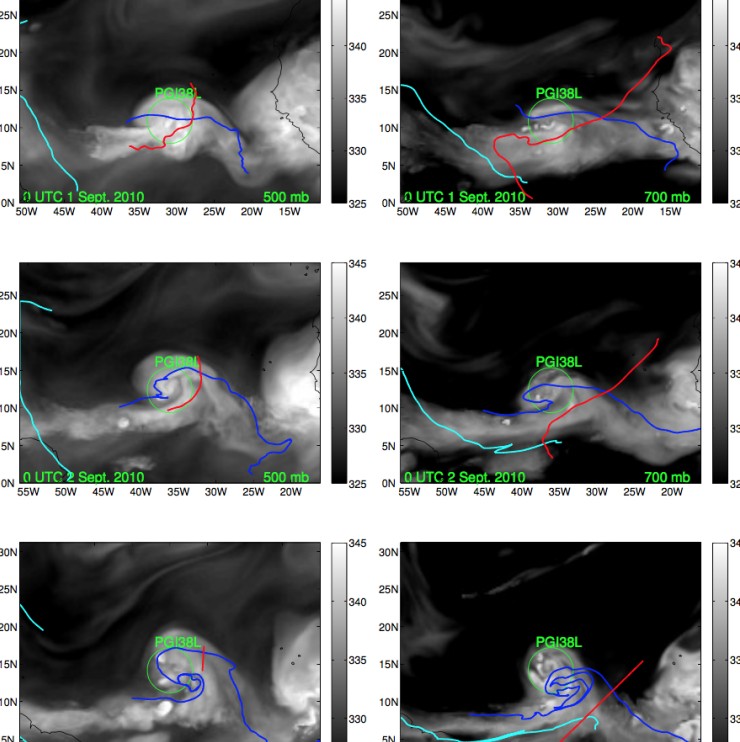

**Figure 6.** Lagrangian manifolds are overlaid on $\theta_e$ fields at 500 hPa (left column) and 700 hPa (right column) from 1 to 3 September. Stable manifolds are red, and unstable manifolds are blue and cyan. The manifolds indicate that the pouch had a hyperbolic point to the east, but was open to environmental air to the west.

Discussion Paper | Discussion Paper | Discussion Paper | Discussion Paper

**ACPD**

doi:10.5194/acp-2015-692

**Why did the storm ex-Gaston (2010) fail to redevelop during the PREDICT experiment?**

T. M. Freismuth et al.

Title Page

Abstract | Introduction

Conclusions | References

Tables | Figures

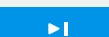 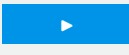

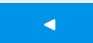 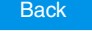

**ACPD**

doi:10.5194/acp-2015-692

**Why did the storm ex-Gaston (2010) fail to redevelop during the PREDICT experiment?**

T. M. Freismuth et al.

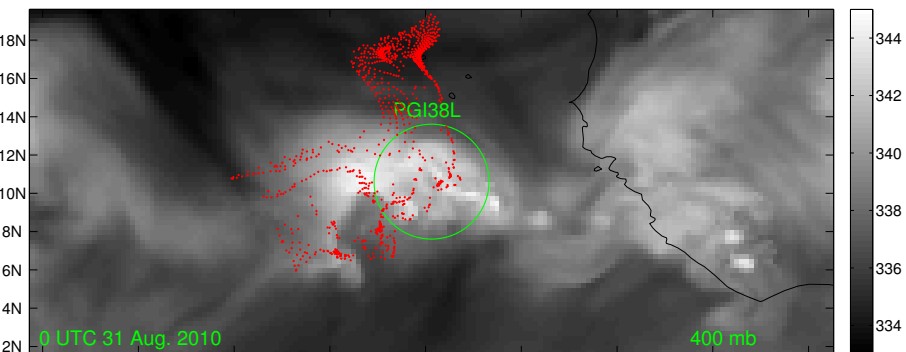

**Figure 7.** Particle trajectory locations at 400 hPa 18:00 UTC 31 August are overlaid on $\theta_e$ (K) valid at 00:00 UTC 31 August 2010. These trajectories are all within a radius of 3° of the pouch center (green circle) by 18:00 UTC 2 September.

**ACPD**

doi:10.5194/acp-2015-692

**Why did the storm ex-Gaston (2010) fail to redevelop during the PREDICT experiment?**

T. M. Freismuth et al.

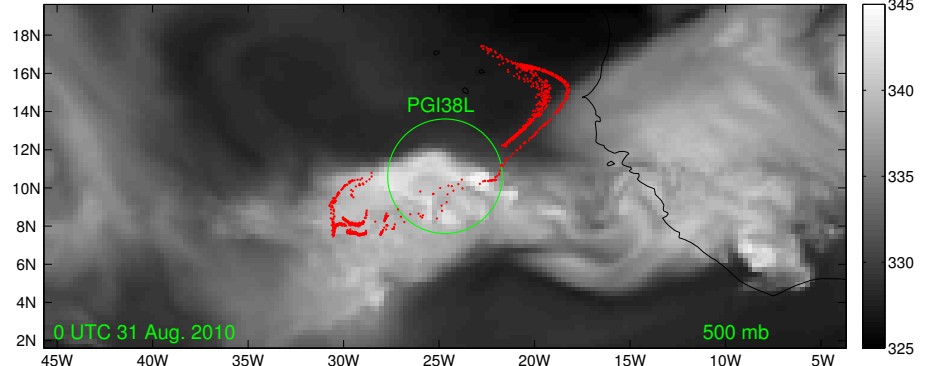

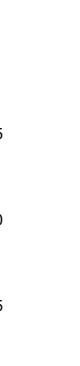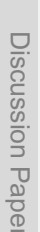

**Figure 8.** Particle trajectory locations at 500 hPa 18:00 UTC 31 August are overlaid on $\theta_e$ (K) 00:00 UTC 31 August 2010. These trajectories are all within a radius of 3° of the pouch center (green circle) by 18:00 UTC 2 September.



# ACPD

doi:10.5194/acp-2015-692

**Why did the storm ex-Gaston (2010) fail to redevelop during the PREDICT experiment?**

T. M. Freismuth et al.

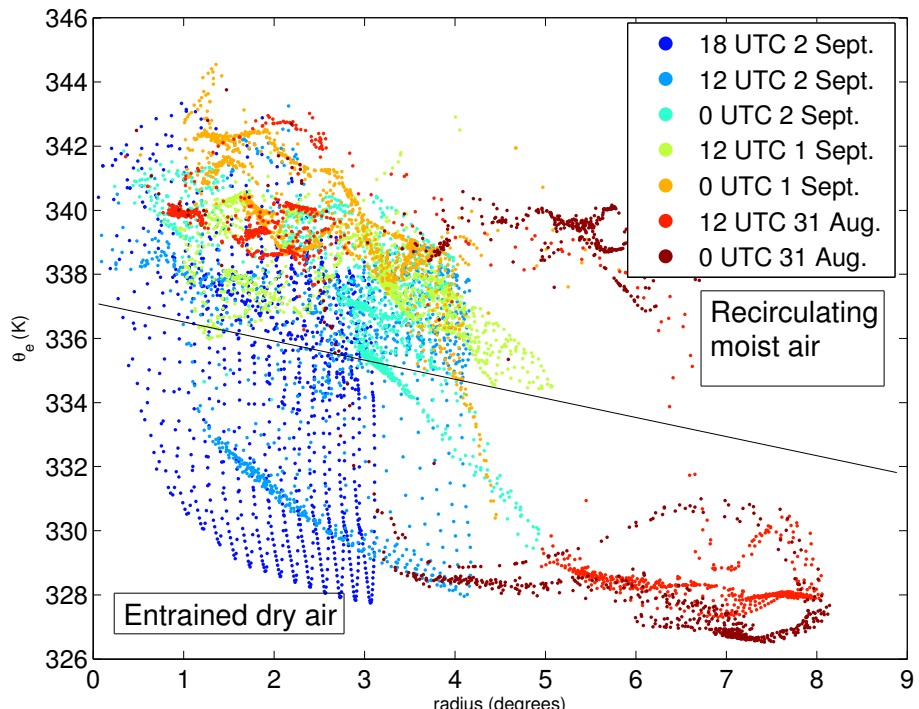

**Figure 9.** $\theta_e$ and radial distance from the center of Gaston's pouch. The colors range from brown to blue, with brown denoting the earliest time of 00:00 UTC 31 August and blue denoting the latest time of 18:00 UTC 2 September.

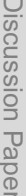

**[ACPD](https://www.atmos-chem-phys-discuss.net)**

doi:10.5194/acp-2015-692

**Why did the storm ex-Gaston (2010) fail to redevelop during the PREDICT experiment?**

T. M. Freismuth et al.

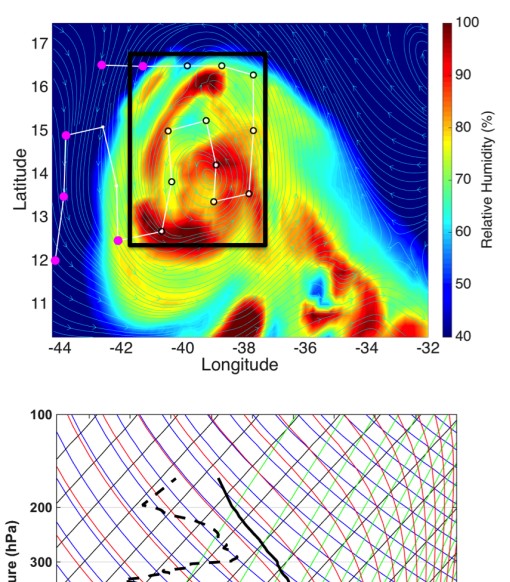

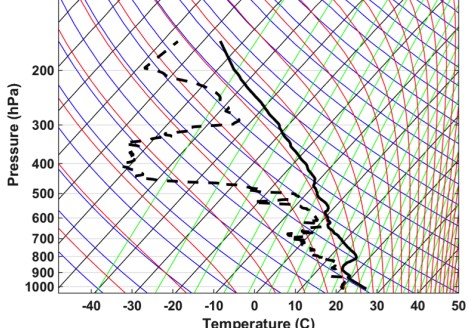

**Figure 10.** PREDICT flight path (white line) and dropsonde locations (dots) overlaid on 700 hPa relative humidity (shading) and co-moving streamlines (cyan lines) from ECMWF analysis data at 18:00 UTC 2 September (top panel). The magenta dots indicate locations where dropsonde data show evidence of an inversion. The black box corresponds to GR14 Fig. 8. Only one of the twelve soundings in the GR14 area of interest shows evidence of an inversion (bottom panel).

# ACPD

doi:10.5194/acp-2015-692

**Why did the storm ex-Gaston (2010) fail to redevelop during the PREDICT experiment?**

T. M. Freismuth et al.

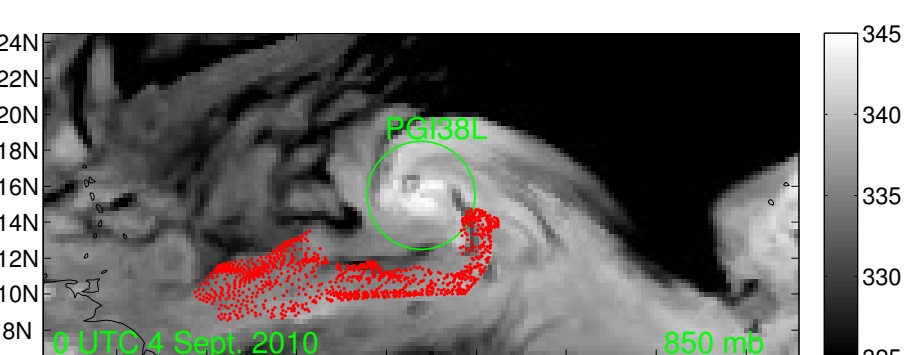

**Figure 11.** Particle trajectory locations at 850 hPa at 00:00 UTC 4 September overlaid on $\theta_e$ (K). Particles were seeded west of ex-Gaston's pouch at 12:00 UTC 2 September, within a suspected trade wind inversion layer. These trajectories are all outside of a 3° radius of the pouch center (green circle) on 4 September.

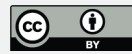