# Peer review of "Why did the storm ex-Gaston (2010) fail to redevelop during the PREDICT experiment?"

_Atmospheric Chemistry and Physics, 2015_

## Referee Comment (RC1) · Anonymous Referee #2 · 15 Feb 2016

Review of

Why did the storm ex-Gaston (2010) fail to redevelop during the PREDICT experiment?

Ref.: acp-2015-692

**Recommendation:** Minor revision

The authors investigate the non development of ex-Gaston using the ECMWF analysis data. By using a dividing streamline methodology, as well as a hyperbolic trajectory methodology, the authors confirm that the "pouch" region was compromised and dry air was able to penetrate. The negative effects of dry air on convective updraughts and vorticty production are described in detail. This manuscript is very polished and I recommend that this study be accepted with some minor revision.

**General comments:**

This is a very useful study that explains coherently why ex-Gaston failed to redevelop. In particular the authors go to great lengths to show that the pouch was not robust, and that dry air was able to penetrate on certain days. Most of my comments are minor issues.

Some of my questions are concerned with the different explanations offered by Gjorgievska and Raymond (2014). With respect to the Lagrangian analysis performed, are there any different conclusions between this study and that of Rutherford and Montgomery (2012)? GR14 state that "Nor did the Lagrangian analysis of Rutherford and Montgomery (2012) show dry air intrusion between the first two Gaston missions."

The analysis of the dropsondes within the 4° box (Fig. 10) shows only a single sounding that shows evidence of a temperature inversion. GR14 claim that they see a temperature inversion in their saturated moist entropy 4° box average. I fail to see how a single sounding with an inversion could influence the box average so strongly. Can you think of possible reasons why there are such discrepancies between your results and those of GR14 in terms of inversions?

**Minor comments:**

- Page 2, lines 23-25. The terms critical layer and critical latitude have not been defined. A definition here is necessary for readers not fimiliar with pouch theory.

- Page 3, lines 12-13. "NASA's ongoing missions …" should be updated to 2016?

- Page 4, line 20. "and NHC", should be "and the NHC".

- Page 5, line 11. How is the pouch center calculated here?

- Page 10, line 23: "Dry air reduced both the updraught and.." I assume you mean updraught strength?

- Page 11, line 13: Can you explain how dry air results in divergence near the 600 hPa level?

- Page 12, line 7-8. "11 of the profiles show evidence of a dry layer above 600 hPa". This sentence needs a "not shown" at the end.

---

## Referee Comment (RC2) · Anonymous Referee #3 · 16 Feb 2016

Why did the storm ex-Gaston (2010) fail to redevelop during the PREDICT experiment?

Freismuth et al. (2016)

Recommendation: Minor revisions

General Comments

Overall, this paper is well written and provides insight into the failed re-development of Tropical Storm Gaston. Although this idea has been examined in several recent studies, the current manuscript provides more insight into the compromise of the protective wave pouch and subsequent intrusion of dry environmental air. The Lagrangian and trajectory analysis illustrate clearly that dry air entered the compromised wave pouch and led to the failed redevelopment of the storm. The authors also provide insight

into the potential role of a trade wind inversion near the wave pouch into the storm evolution.

Specific Comments

P5, Line 24-26: What is the significance/importance of the deep shear changing direction from NE-SW-E over a short period on 2 Sep?

P8, Line 6: There is no analysis of the kinematic structure of the wave pouch at 400 hPa, but particle trajectories are examined in Fig. 7. A simple statement describing its structure should be added.

P17, Table 1: It would be nice to somehow indicate what dropsondes are within the three degree box you use for analysis and which are outside.

P23, Fig. 6: Is the unstable manifold represented by the cyan line a feature of the Gaston wave pouch, or some larger scale feature?

Technical Corrections

P2, Line 9: Omit comma after vertical

P6, Line 9: "referred to as pouch scales" should be used on the previous page to explain why you chose the three-degree box for your analysis domain.

P7, Line 3: First mention of RM12, needs to be defined. It would be nice to distinguish in the beginning the differences between the present study and the work of RM12.

P9, Line 1: Degree symbol after 0.

P10, Line 7: Are the listed RH & CAPE values equal to or less than?

---

## Referee Comment (RC3) · D. Raymond (Referee) · 23 Feb 2016

**Review of "Why did the storm ex-Gaston (2010) fail to redevelop during the PREDICT experiment?" by Freismuth et al.**

David J. Raymond

February 14, 2016

The authors of this manuscript make parcel trajectory calculations for the failed tropical cyclone Gaston (2010), based on ECMWF analyses that included dropsonde data from the PREDICT project. There is general agreement that Gaston decayed after the second PRE-DICT mission on 3 September 2010 as a result of ingestion of dry air. However, the current manuscript as well as a couple of other papers cited by the authors further assert that the decay of Gaston between 2 and 3 September was also due to the incorporation of dry air.

Gjorgjievska and Raymond (2014; GR2014, cited in the manuscript) do not dispute that dry air was instrumental in the decay of Gaston after 3 September. However we also demonstrate that a more subtle process was likely occurring in the 2-3 September interval that led to the subsequent flood of dry air invading Gaston.

The first hint that dry air did not affect the convection in Gaston prior to 3 September comes from in figure 3d of GR2014. (Note that due to an unfortunate transposition error, the images for figures 3 and 4 are switched, so the image for figure 3 is shown with the figure 4 caption.) This figure demonstrates that the relative humidity averaged over a 4 by 4 degree box centered on roughly on the 5 km vortex center changed very little between 2 and 3 September. The main difference is an *increase* in the relative humidity near the 5 km vortex center in the 7-9 km range between these two dates. (See also figure 5 of GR2014).

Figure 6 of the current manuscript shows the analyzed equivalent potential temperature at 500 hPa (approximately 5 km) on 1-3 September. There is indeed a dry tendril of air sweeping around the south and east side of Gaston 1 and 2 (on 2 and 3 September respectively), but the actual 5 km circulation centers in the co-moving frame were at (39W, 15N) and (42W, 15N) on these two days, i.e., 2-4 degrees to the north and west of the dry air intrusions (see GR2014 figure 5).

Figure 6 of GR2014 shows the vertical mass flux pattern at 700 hPa in Gaston 1 on 2 September. The strongest upward motion (representing deep convection) is centered at (39.5W,14.5N), or slightly to the SW of the 500 hPa circulation center. There is evidence of downward motion roughly 1.5-2 degrees to the east of this ascent, possibly representing the effects of the intruding dry air. Nevertheless, convection responds to the thermodynamics of air in its immediate vicinity, not to air 150-200 km away, indicating that the convective core of Gaston on this date was still narrowly protected from dry air by the pouch.

Comparison of Gaston 1 and 2 with the developing cyclone Karl shows that the relative humidity profiles in the early stages of Karl were very similar to that of Gaston 2. Yet Karl developed into a major hurricane. The most obvious difference between the two cases is that Gaston 1 and 2 experienced SSTs of 28.2 C and 28.4 C respectively, whereas the first 3 Karl missions showed SSTs of 30-30.2 C (see table 1 of GR2014). Thus the SSTs in the early Karl stages exceeded those in Gaston by almost 2 C. In addition, as figure 9 of GR2014 shows, the tropical cyclone heat potential for Gaston was quite small in its initial stages and quite large for Karl.

Figure 10a of GR2014 shows that the environment of convection in Gaston 1 had strong convective inhibition near 2 km, and that considerable energy had to have been expended by the convection in breaking this inhibiting layer in the convective region. Such an inhibiting layer did not exist in the vicinity of convection in Karl 3 (11 September 2010), as figure 10c shows. The inhibiting layer in Gaston relative to Karl is almost certainly related to the lower SST experienced by Gaston.

As Figure 7 of GR2014 shows, the convective mass flux profile for Karl 3 was vastly different from that of Gaston 1, with extreme top-heavy convection in Karl 3 and extreme bottom-heavy convection in Gaston 1. This resulted in much stronger convergence below 2 km and a corresponding increase in the strength of the low-level circulation between Gaston 1 and Gaston 2 (see figure 3a – shown under the figure 4 caption as noted above). However, strong divergence above 3 km in Gaston 1 resulted in the destruction of an initially strong mid-level vortex, as figure 3a shows.

GR2014 argue that the elimination of the mid-level vortex weakened the pouch sufficiently to allow the ingestion of dry air, resulting in the subsequent decay of Gaston. Given that the relative humidity profiles for Gaston 1 and 2 and Karl 3 were nearly identical, as are the parcel buoyancy profiles above 3 km, the existence of strong convective inhibition in the environment of Gaston 1, undoubtedly related to the lower SST, is the most plausible explanation for the dramatic differences between the convection in the two cases. (As noted by the authors of this manuscript, the most extreme convective inhibition, as represented by a trade wind inversion, occurred well to the west of Gaston 1. However, relatively strong convective inhibition, as noted above, existed on all sides of the convective core in this case.)

In summary, the evidence for our view of the decay of Gaston before 3 September consists of 2 parts: (1) The relative humidity did not decrease and in fact increased at upper levels between Gaston 1 and Gaston 2 in a region centered on the 5 km circulation center. The convective core was very close to the circulation center in these two cases. (2) The low SSTs and increased static stability near the convective core of Gaston likely had a negative effect on convection in Gaston 1 even if there was technically no trade wind inversion in the convectively active area. This stands out particularly in comparison to Karl 3, in which convective inhibition was weak over the entire region, and for which the SSTs were much higher.

Part of the discrepancy between the results of GR2014 and the current manuscript may be due to the location of the pouch. For both Gaston 1 and Gaston 2, the 5 km circulation

centers are on the NW edge of the pouch positions as defined in the manuscript (see figures 5a and 5b in GR2014 in comparison with figure 6 in the manuscript). Furthermore, the convective cores in these cases are much closer to the 5 km circulation centers than to the center of the pouches defined in the manuscript under review (see figure 6a in GR2014 for Gaston 1; Gaston 2 not shown). One can of course define the pouch in accordance with the circulation center at any level one desires, and Montgomery and colleagues tend to define this at 850 hPa (or perhaps 700 hPa in this paper – this is not clear). For reasons set forth in Raymond et al. (2014; Tropical cyclogenesis and mid-level vorticity. Australian Meteorological and Oceanographic Journal, 64, 11-25.) we prefer a higher level, i.e., near 5 km in many cases. Given that the convection tends to occur near the 5 km circulation center on both days, the higher level would seem to be more appropriate in this case.

The dependence on a global analysis for very delicate Lagrangian trajectory calculations also raises at least a yellow flag. Analyses incorporate sounding data in competition with model prejudices with opaque weighting factors. Our analyses depend on PREDICT dropsonde data only.

I feel that I am perhaps too close to this whole argument to give an objective recommendation on this paper, so I shall leave that to the other reviewers and the editor. However, though I do appreciate the authors' attempt to represent our position in their manuscript, I would like to see the whole story told, which explains the length of this commentary. Technically, the manuscript is well written, though some of the figures, such as figure 5, are very hard to decipher.

---

## Author Response (AR1)

**Response to Reviewer 1:**

With respect to the Lagrangian analysis performed, are there any different conclusions between this study and that of Rutherford and Montgomery (2012)? GR14 state that "Nor did the Lagrangian analysis of Rutherford and Montgomery (2012) show dry air intrusion between the first two Gaston missions."

Our analysis of the dry air in this study is more explicit, with trajectories and manifolds computed directly and moisture analyzed along the trajectories. In contrast, RM12 analyzed the dry air intrusion more implicitly, via a tracer field, that would still have allowed for modification of the air during entrainment.

Can you think of possible reasons why there are such discrepancies between your results and those of GR14 in terms of inversions?

We are reluctant to speculate here. We think our analysis provides a picture that is consistent with the analyzed Lagrangian flow in and around Gaston's weak pouch. When one compares our results with GR14, it seems clear that we invoke a simpler dynamical interpretation of the data using new insights on convective-vorticity dynamics obtained by Kilroy and Smith concerning the negative impact of dry air on vorticity amplification.

Page 2, lines 23-25. The terms critical layer and critical latitude have not been defined. A definition here is necessary for readers not familiar with pouch theory.

We have added the requested definitions in the revised text.

Page 3, lines 12-13. "NASA's ongoing missions …" should be updated to 2016?

We have revised the text to include 2016.

Page 4, line 20. "and NHC", should be "and the NHC".

We corrected the error in the revised text.

Page 5, line 11. How is the pouch center calculated here?

The pouch center is defined as the intersection of the trough and critical latitude at the 700 hPa level. This has been clarified in the revised text.

Page 10, line 23: "Dry air reduced both the updraught and.." I assume you mean updraught strength?

Yes, the implication is that updraught strength is reduced. This has been clarified in the revised manuscript.

Page 11, line 13: Can you explain how dry air results in divergence near the 600 hPa level?

The reviewer is correct that *dry air, itself, does not cause divergence*. For the paragraph in question, we were attempting to link the work of KS12 to the elevated dry air observed in the PREDICT soundings.

The ECMWF analyses (with PREDICT data) indicate that there is an upward mass flux, that decreases in magnitude with height, up to the dry, 600 hPa level. From the continuity equation, one can see that a decrease of mass flux at and above the 600 hPa level implies a horizontal divergence of mass at these levels.

We have revised the text in question to clarify our scientific argument:

"Based on the findings of KS12, convective updraughts that form in this region containing dry air aloft would be expected to result in divergence near the 600 hPa level, thus causing an expanding material loop at these levels."

Page 12, line 7-8. "11 of the profiles show evidence of a dry layer above 600 hPa". This sentence needs a "not shown" at the end.

We made this correction in the revised text.

**Response to reviewer 2:**

P5, Line 24-26: What is the significance/importance of the deep shear changing direction from NE-SW-E over a short period on 2 Sep?

We are simply describing the local flow environment and how it evolved during the period of observations. We do not have evidence or theoretical intuition to suggest that the change in direction is dynamically or thermodynamically significant. Consequently, we do not feel a remark is needed in the text.

P8, Line 6: There is no analysis of the kinematic structure of the wave pouch at 400 hPa, but particle trajectories are examined in Fig. 7. A simple statement describing its structure should be added.

Manuscript text updated to discuss the 400 hPa wave pouch.

P17, Table 1: It would be nice to somehow indicate what dropsondes are within the three degree box you use for analysis and which are outside.

Annotations were added to the table in the revised manuscript.

P23, Fig. 6: Is the unstable manifold represented by the cyan line a feature of the Gaston wave pouch, or some larger scale feature?

Technical Corrections
P2, Line 9: Omit comma after vertical

We corrected this error in the revised manuscript.

P6, Line 9: "referred to as pouch scales" should be used on the previous page to explain why you chose the three-degree box for your analysis domain.

We made this edit to the updated text.

P7, Line 3: First mention of RM12, needs to be defined. It would be nice to distinguish in the beginning the differences between the present study and the work of RM12.

We corrected this error with an inline citation. References have been updated also in the revised text.

P9, Line 1: Degree symbol after 0.

We made this correction in the updated text.

P10, Line 7: Are the listed RH & CAPE values equal to or less than?

The listed RH and CAPE values are "equal to".

**Response to review by D. Raymond:**

The authors of this manuscript make parcel trajectory calculations for the failed tropical cyclone Gaston (2010), based on ECMWF analyses that included dropsonde data from the PREDICT project. There is general agreement that Gaston decayed after the second PREDICT mission on 3 September 2010 as a result of ingestion of dry air. However, the current manuscript as well as a couple of other papers cited by the authors further assert that the decay of Gaston between 2 and 3 September was also due to the incorporation of dry air.

Gjorgjievska and Raymond (2014; GR2014, cited in the manuscript) do not dispute that dry air was instrumental in the decay of Gaston after 3 September. However we also demonstrate that a more subtle process was likely occurring in the 2-3 September interval that led to the subsequent flood of dry air invading Gaston.

The first hint that dry air did not affect the convection in Gaston prior to 3 September comes from in figure 3d of GR2014. (Note that due to an unfortunate transposition error, the images for figures 3 and 4 are switched, so the image for figure 3 is shown with the figure 4 caption.) This figure demonstrates that the relative humidity averaged over a 4 by 4 degree box centered on roughly on the 5 km vortex center changed very little between 2 and 3 September. The main difference is an increase in the relative humidity near the 5 km vortex center in the 7-9 km range between these two dates. (See also figure 5 of GR2014).

**With due respect, just because the area-averaged RH changes little on the 4 by 4 degree box moving with the system does not unequivocally imply that the inner pouch was isolated and protected from its environment. As an example, the convection that was observed during Gaston 1 would be expected to moisten the middle levels inside the pouch; if dry air was intruding into the system (which we show to be the case in Figure 6), this convective moistening could work to offset the dry air entrainment. Given the demonstrated intrusion of dry air into the pouch, we would expect the convection to be negatively impacted by the dry air according to the findings of Kilroy and Smith (2012).**

Figure 6 of the current manuscript shows the analyzed equivalent potential temperature at 500 hPa (approximately 5 km) on 1-3 September. There is indeed a dry tendril of air sweeping around the south and east side of Gaston 1 and 2 (on 2 and 3 September respectively), but the actual 5 km circulation centers in the co-moving frame were at (39W, 15N) and (42W, 15N) on these two days, i.e., 2-4 degrees to the north and west of the dry air intrusions (see GR2014 figure 5).

**We believe we can explain the apparent discrepancy of the circulation center as a misunderstanding of the timing of the data and the appropriate figures. The dropsondes during the first flight into Gaston (GR2014's Gaston 1) were deployed during 1532-1906 UTC on 2 September, which is closer to 0000 UTC 3 September rather than 2 September. The Reviewer uses our 2 September plots (Fig. 6, middle)**

**to compare with his Gaston 1 data, but our 3 September plots (Fig. 6, bottom) are actually more appropriate.  The GR2014 Gaston 1 position of (39W, 15N) actually agrees well with our 3 September Fig. 6 (bottom) plots.  It should be noted that the Gaston 2 flight that occurred late on 3 September (closer to 4 September) is not represented by any of our figures.  However, upon inspection of our other analyses not included in this manuscript, we believe that our pouch positions are actually in agreement with the Reviewer on that later flight as well.  Considering the circulation's WNW motion, GR2014's Gaston 2 position of (42W, 15N) based upon dropsondes deployed during 1444-1849 UTC 3 September corresponds fairly well with the subsequent ECMWF 0000 UTC 4 September pouch position of about (43.5W, 15.5N).**

**Bottom line: There is no 2 – 4 degree positioning error of the pouch center as suggested by the reviewer.**

Figure 6 of GR2014 shows the vertical mass flux pattern at 700 hPa in Gaston 1 on 2 September. The strongest upward motion (representing deep convection) is centered at (39.5W, 14.5N), or slightly to the SW of the 500 hPa circulation center. There is evidence of downward motion roughly 1.5-2 degrees to the east of this ascent, possibly representing the effects of the intruding dry air. Nevertheless, convection responds to the thermodynamics of 1 air in its immediate vicinity, not to air 150-200 km away, indicating that the convective core of Gaston on this date was still narrowly protected from dry air by the pouch.

**GR2014 recognize that asymmetries may alter averages, saying that any way of averaging may give inconsistent results.  In at least two places in their mss., they recognize the role that time-dependent non-linear dynamics could have played.  The first is when they note that the increase in the mid-level vorticity went against a negative vorticity tendency.  The second, is when they note that transport of dry air could have been a factor in the decay of Gaston, however, the pouch (according to GR2014) was apparently closed.  The difference in sign between vorticity tendency and actual vorticity evolution is a very strong indication that time-dependent dynamics plays a role.  Our study here uses the Lagrangian manifolds as a way to measure the role of time-dependent dynamics objectively, that is, without any sensitivity to the choice of spatial location in averaging.**

Comparison of Gaston 1 and 2 with the developing cyclone Karl shows that the relative humidity profiles in the early stages of Karl were very similar to that of Gaston 2. Yet Karl developed into a major hurricane. The most obvious difference between the two cases is that Gaston 1 and 2 experienced SSTs of 28.2 C and 28.4 C respectively, whereas the first 3 Karl missions showed SSTs of 30-30.2 C (see table 1 of GR2014). Thus the SSTs in the early Karl stages exceeded those in Gaston by almost 2 C. In addition, as figure 9 of GR2014 shows, the tropical cyclone heat potential for Gaston was quite small in its initial stages and quite large for Karl.

**The GR2014 argument appears to be that Karl had 30 C SSTs and Gaston had 28.5 C SSTs, therefore Gaston did not develop because it had greater *convective inhibition* than Karl. This is a case where the models predicted the SSTs fairly well, unlike Nate (2011) where the upwelling effect of the storm led to errors in the SSTs, yet the models had no trouble predicting development.**

**It seems highly plausible that there is a feedback between less convection and greater permeability of the pouch boundary, since the vorticity gradients caused by convection and its associated convergence determine the strength of the boundary. In other words, a weaker boundary leads to dry air intrusions, which further weakens the convection.**

**GR2014 do not explicitly consider the aforementioned feedback between dry air intrusion and weakening of convection. Their focus is rather on the supposition that the SSTs played an essential role, and only then because Karl happened to have higher SSTs and did develop.**

**Our study has focused on the proposed potential feedback effect and how a weaker boundary further hinders development. This line of study is different from the GR2014 line (and the reviewer's line above) that lower SSTs underneath ex-Gaston's pouch were the most obvious difference between it and the developing Karl. It is noteworthy to point out that in numerical cyclogenesis studies, an SST of 28.5C is not subcritical provided, inter alia, the initial vorticity is favorable. From a larger-scale perspective, however, the mid-level vorticity for Gaston was unfavorable; there was significant vertical shear. We can't absolutely say that the vertical shear would or would not have broken the pouch boundary if convection had been stronger. What we do show is that dry air intruded, and the dry air was not moistened until it entered the pouch. That suggests that the dry air played at least some role in limiting the convection.**

**Summary: GR2014 do show ocean tropical cyclone heat potential, in addition to noting the minor SST difference between ex-Gaston and Karl. We acknowledge that there is a potential feedback between the ocean heat content and dynamics of the pouch boundary, but it seems unreasonable to dismiss lateral intrusions when: i) many of GR2014's results are possibly artifacts of spatial averaging and ii) the actual dynamics suggest that Lagrangian versus Eulerian temporal sampling are so different.**

Figure 10a of GR2014 shows that the environment of convection in Gaston 1 had strong convective inhibition near 2 km, and that considerable energy had to have been expended by the convection in breaking this inhibiting layer in the convective region. Such an inhibiting layer did not exist in the vicinity of convection in Karl 3 (11 September 2010), as figure 10c shows. The inhibiting layer in Gaston relative to Karl is almost certainly related to the lower SST experienced by Gaston.

**Convective inhibition (and the corresponding acronym `CIN') is mentioned only**

once in the GR2014 mss. on page 3068 (section 4, right column, middle paragraph) in a Background section reviewing Raymond and Sessions 2007 and its argued application to the real world.

On that page of the GR2014 mss., CIN is mentioned in the context of 'parcel buoyancy' below 2 km altitude. The term `parcel buoyancy' is used four times in this paragraph and this is the only place in their mss. where the term parcel buoyancy is mentioned. (Parcel buoyancy is not defined mathematically in the GR2014 mss.)

CIN is never used in the GR2014 mss. to compare Gaston and Karl. Rather, the non-standard 'instability index' (their Eq. (6), same page, left column) is used as a measure of a *system instability (our words and emphasis)*. Logic suggests that one should not use a non-standard definition of 'CIN' to compare against the CIN from another study using a standard definition.

Traditionally, CIN is defined as the work done required to lift a moist air parcel to its level of free convection (e.g., Emanuel 1994). Since this work depends on the parcel, one needs to take a further step and make the definition unambiguous. In Smith and Montgomery (2012, QJRMS; hereafter SM12), CIN was defined and calculated using the *minimum work required to lift a test parcel to its level of free convection*. Although the minimum work usually corresponds to parcels lifted from the surface, this is not always the case. SM12 calculated CIN defined in this (standard) way for Gaston, Karl and Matthew pre-storms. In particular, the CIN calculated for ex-Gaston on 2 September is shown in their Figure 6 and the CIN calculated for both flights into pre-Karl on 10 September are shown in their Figure 10. After carefully examining the data plotted in these figures, one does not find that the CIN for ex-Gaston is larger than that for Karl. In fact, in the vicinity of the sweet spot, one finds that the opposite is true! As an example, we calculate below the arithmetic average of CIN values within a horizontal radius of approximately 2 degrees from the center of the ECMWF-analyzed sweet spot (indicated by the green curve) in SM12's Figure 6 and the top right plot of SM12's Figure 10 (first flight of Karl). (As a point of clarification, the 2 degree radius circle corresponds roughly with the 4 by 4 degree box averages used by GR2014.)

We find the following results:

Ex-Gaston Average CIN within 2 degrees radius from analyzed sweet spot: $9 + 29 + 0 + 11 + 0 + 0 + 5 + 58$ J kg$^{-1}$ / 8 = 112 J kg$^{-1}$/8 = 8 J Kg$^{-1}$

Karl 1 Average CIN within 2 degrees radius from analyzed sweet spot: $4 + 18 + 40 + 16 + 14 + 7 + 47 + 36 + 21$ J kg$^{-1}$ / 9 = 203 J kg$^{-1}$/9 = 22.2 J Kg$^{-1}$

Karl 1 CIN is more than a factor of two larger than that for ex-Gaston.

A similar result is found for Karl 2 (the second GV flight on 10 September).

**Karl 2 Average CIN within 2 degrees radius from analyzed sweet spot: 37 + 11 + 33 + 20 + 29 + 1 + 24 + 9 + 33 + 16 J kg^{-1} / 10 = 213 J kg^{-1}/10 = 21.3 J Kg^{-1}**

**Admittedly, we have chosen a 2 degree radius around the sweet spot and we could have chosen a larger radius or a smaller radius. But the results will not change significantly so as to render ex-Gaston CIN > Karl 1,2 CIN.**

**Thus, based on the evidence presented in Smith and Montgomery (2012), the hypothesis that ex-Gaston did not develop because its CIN was larger than that of Karl is rejected. Our finding here is in accord with one of the primary conclusions of Smith and Montgomery (p1738) who stated that:**

**"Even so, the evolution and distribution of CAPE and CIN by themselves did not reveal an obvious distinction between developing and non-developing systems."**

As Figure 7 of GR2014 shows, the convective mass flux profile for Karl 3 was vastly different from that of Gaston 1, with extreme top-heavy convection in Karl 3 and extreme bottom-heavy convection in Gaston 1. This resulted in much stronger convergence below 2 km and a corresponding increase in the strength of the low-level circulation between Gaston 1 and Gaston 2 (see figure 3a – shown under the figure 4 caption as noted above). However, strong divergence above 3 km in Gaston 1 resulted in the destruction of an initially strong mid-level vortex, as figure 3a shows. GR2014 argue that the elimination of the mid-level vortex weakened the pouch sufficiently to allow the ingestion of dry air, resulting in the subsequent decay of Gaston. Given that the relative humidity profiles for Gaston 1 and 2 and Karl 3 were nearly identical, as are the parcel buoyancy profiles above 3 km, the existence of strong convective inhibition in the environment of Gaston 1, undoubtedly related to the lower SST, is the most plausible explanation for the dramatic differences between the convection in the two cases. (As noted by the authors of this manuscript, the most extreme convective inhibition, as represented by a trade wind inversion, occurred well to the west of Gaston 1. However, relatively strong convective inhibition, as noted above, existed on all sides of the convective core in this case.)

**For the reasons given above, with due respect we do not accept the loose association of lower SSTs and convective inhibition and we think one should not use non-standard definitions of 'CIN' to compare against the CIN from another study using a standard definition. Based on the calculations presented by SM12, and the additional calculations summarized above, we do not find that the dropsonde data supports the reviewer's assertion that the "convective inhibition" was relatively strong on all sides of Gaston 1.**

In summary, the evidence for our view of the decay of Gaston before 3 September consists of 2 parts: (1) The relative humidity did not decrease and in fact increased at upper levels between Gaston 1 and Gaston 2 in a region centered on the 5 km circulation center. The convective core was very close to the circulation center in these two cases. (2) The low SSTs and increased static stability near the convective core of Gaston likely

had a negative effect on convection in Gaston 1 even if there was technically no trade wind inversion in the convectively active area. This stands out particularly in comparison to Karl 3, in which convective inhibition was weak over the entire region, and for which the SSTs were much higher.

Part of the discrepancy between the results of GR2014 and the current manuscript may be due to the location of the pouch. For both Gaston 1 and Gaston 2, the 5 km circulation centers are on the NW edge of the pouch positions as defined in the manuscript (see figures 5a and 5b in GR2014 in comparison with figure 6 in the manuscript). Furthermore, the convective cores in these cases are much closer to the 5 km circulation centers than to the center of the pouches defined in the manuscript under review (see figure 6a in GR2014 for Gaston 1; Gaston 2 not shown). One can of course define the pouch in accordance with the circulation center at any level one desires, and Montgomery and colleagues tend to define this at 850 hPa (or perhaps 700 hPa in this paper – this is not clear). For reasons set forth in Raymond et al. (2014; Tropical cyclogenesis and mid-level vorticity. Australian Meteorological and Oceanographic Journal, 64, 11-25.) we prefer a higher level, i.e., near 5 km in many cases. Given that the convection tends to occur near the 5 km circulation center on both days, the higher level would seem to be more appropriate in this case.

**The nominal center depicted in this mss. is defined as the intersection between the wave trough and the local critical curve at the tracking level (700 mb). We have clarified this point in the revised mss.**

**The apparent discrepancy between the GR2014 pouch positions and our sweet spot positions has been largely resolved by our response to the reviewer above.**

The dependence on a global analysis for very delicate Lagrangian trajectory calculations also raises at least a yellow flag. Analyses incorporate sounding data in competition with model prejudices with opaque weighting factors. Our analyses depend on PREDICT dropsonde data only.

**While individual trajectory computations are sensitive to the trajectory integration scheme and the quality of the wind data, Lagrangian coherent structure identification is surprisingly robust (Haller 2002, DOI:10.1063/1.1477449). This study provides a detailed analysis of the wind fields, which over any finite time interval must satisfy momentum conservation within the model. Therefore, for the purposes at hand we feel that the model wind fields provide a better depiction of time-dependent velocities than instantaneous wind fields derived from dropsonde data. As an additional affirmation of the consistency of our methodology employed here, the Lagrangian manifolds that we have computed are in agreement with both the model moisture fields and vorticity fields.**

I feel that I am perhaps too close to this whole argument to give an objective recommendation on this paper, so I shall leave that to the other reviewers and the editor. However, though I do appreciate the authors' attempt to represent our position in their

manuscript, I would like to see the whole story told, which explains the length of this commentary. Technically, the manuscript is well written, though some of the figures, such as figure 5, are very hard to decipher.

**We thank the reviewer for his careful reading of the manuscript, and for his perceptive and thorough review. For the reasons given in our manuscript and in our responses above, we have a very different interpretation of the failed development of Gaston (before 03 September).  In our study, we show (1) the pouch is open and vulnerable as early as 1 September (Figure 6), (2) dry environmental air was entrained into the pouch (Figures 7 and 8) as early as 1 September, and (3) vorticity and vertical mass flux decrease with decreasing relative humidity (Figure 4).**

**We think the data supports the foregoing dynamical interpretation that builds on the new insights by Kilroy and Smith (2012) concerning the negative impact of dry air on vorticity amplification within the pouch of pre-storm disturbances.**

Manuscript prepared for Atmos. Chem. Phys. Discuss.
with version 2015/04/24 7.83 Copernicus papers of the LATEX class copernicus.cls.
Date: 15 May 2016

**Why did the storm ex-Gaston (2010) fail to redevelop during the PREDICT experiment?**

**T. M. Freismuth**[1], **B. Rutherford**[2], **M. A. Boothe**[1], **and M. T. Montgomery**[1]

[1]Naval Postgraduate School, Monterey, CA, USA
[2]Northwest Research Associates, Redmond, WA, USA

Correspondence to: M. T. Montgomery (mtmontgo@nps.edu)

**Abstract**

An analysis is presented of the failed re-development of ex-Gaston during the 2010 PRE-DICT field campaign based on the European Centre for Medium Range Weather Forecast (ECMWF) analyses. We analyze the dynamics and kinematics of ex-Gaston to investigate the role of dry, environmental air in the failed redevelopment. The flow topology defined by the calculation of particle trajectories shows that ex-Gaston's pouch was vulnerable to dry, environmental air on all days of observations. As early as 12:00 UTC 2 September 2010, a dry layer at and above 600 hPa results in a decrease in the vertical mass flux and vertical  relative vorticity. These findings support the hypothesis that entrained, dry air near 600 hPa thwarted convective updraughts and vertical mass flux, which in turn led to a reduction in vorticity and a compromised pouch at these middle levels. A compromised pouch allows further intrusion of dry air and  quenching of subsequent convection, therefore hindering vorticity amplification through vortex tube stretching. This study supports recent work investigating the role of dry air in moist convection during tropical cyclogenesis.

**1 Introduction**

Recent work has established a new overarching framework for understanding tropical cyclone formation from easterly waves (Dunkerton et al., 2009, hereafter DMW09). This framework, for describing how such hybrid wave-vortex structures develop into tropical depressions, was likened to the development of a marsupial infant in its mother's pouch. By analogy, a juvenile proto-vortex is carried along by its parent wave until the proto-vortex is strengthened into a self-sustaining entity. For tropical storms developing from within tropical waves, the recirculating flow in the wave's critical layer corresponds to the "wave-pouch". Here, the wave and mean-flow speeds are similar, along a critical latitude oriented approximately parallel to the easterly jet, and the trough axis intersects meridionally. The critical

latitude is the latitude where the mean flow and wave phase speeds are equal (DMW09). Storm formation occurs typically near the intersection of critical latitude and trough axis.[1]

The new cyclogenesis model and accompanying scientific hypotheses were established observationally in the Atlantic and eastern Pacific sectors by DMW09. They find additional

5 support in idealized numerical modeling simulations (Wang et al., 2010a, b; Montgomery et al., 2010b; Nicholls and Montgomery, 2013), recent case studies in the field in the western North Pacific during the Tropical Cyclone Structure Experiment 2008 (TCS08, Montgomery et al., 2010a; Lussier III, 2010; Montgomery et al., 2012; Raymond and Lopez-Carrillo, 2011; Lussier III et al., 2014), in the Atlantic during the Pre-Depression Investi-

10 gation of Cloud Systems in the Tropics (PREDICT) campaign in 2010 (Montgomery et al., 2012; Smith and Montgomery, 2012; Davis and Ahijevych, 2012, 2013), in NASA's ongoing Hurricane and Severe Storm (HS3) missions (2012–2015 2012–2016) and the case of Hurricane Sandy (Lussier III et al., 2015). The field data afford a resolved view of horizontal and vertical structure in the wave pouch and its immediate surroundings, valuable for

15 system centering, circulation magnitude, vorticity balance, interleaving of air masses, and moist thermodynamic profiles.

A corollary from the new model is that the non-development of a candidate tropical disturbance is linked to the pouch structure being compromised. Currently, it is thought there are two principal ways the pouch can be compromised. The first way is a kinematic

20 combined kinematic-dynamic effect caused by the differential shearing of the pouch in the vertical plane or horizontal plane, or both. Shearing . The increased shear and deformation of the pouch degrades the protective womb that both nurtures the incipient proto-vortex and supports deep convective activity tends to compromise the resilience of the vortex (Reasor et al., 2004) and produce a vertically misaligned distribution of moisture generated

25 by the convection. The second way is a combined thermodynamic-dynamic effect associated with the intrusion of dry air (so-called "anti-fuel") into the otherwise moist pouch from
* * *
[1]The jet contains two such critical latitudes, the cyclonic one equatorward of the jet axis being instrumental to storm formation, the anticyclonic one poleward of the jet axis relevant to dusty Saharan air outbreaks and dry subsidence aloft.

a relatively dry environment. The injection of anti-fuel into the  wave-pouch acts to limit the vigor of deep convection  in the middle and upper troposphere and the amplification of vertical vorticity in convective updraughts above the boundary layer (Kilroy and Smith, 2012), which is essential for spinning up a tropical cyclone (Smith and Montgomery, 2012).

The non-developing case of ex-Gaston (2010) during the PREDICT experiment is arguably one of the most extensively observed non-developing tropical disturbances ever. The five consecutive days of observational data for such a non-developing disturbance is unprecedented.

Based on the foregoing discussion, there remains an important question in understanding the non-development of ex-Gaston: Did ex-Gaston have a robust (closed), protective pouch? If ex-Gaston did, in fact, have a robust pouch, one would expect the system to redevelop and possibly intensify. We will show that ambient vertical shear and the entrainment of dry, environmental air early on 2 September led to the degradation of ex-Gaston's pouch and this plagued the convection within the pouch for the entire observational period of the PREDICT experiment.

**2 Review of Pre-PREDICT Gaston**

Tropical Storm Gaston developed from an African easterly wave that moved westward from the African coast on 28 August 2010. The National Hurricane Center (NHC) designated Gaston as a tropical storm at 15:00 UTC 1 September. Despite being in a favorable environment with relatively low vertical shear (discussed further below) and a SST of 28.5 °C (Gjorgjievska and Raymond, 2014), convection associated with Gaston quickly diminished, and the NHC downgraded the system to a post-tropical/remnant low by 21:00 UTC 2 September. Convective activity increased on 3 September, however it did not re-organize and the system remained a remnant low.

**3 Data sources**

This study uses the European Centre for Medium-Range Weather Forecasts (ECMWF) analyses from 28 August 2010 to 11 September 2010. The analysis fields have a horizontal resolution of 0.25°, 25 vertical levels from 1 to 1000 hPa, and temporal output every 6 h.

5 Dropsonde data from the Pre-Depression Investigation of Cloud-Systems in the Tropics (PREDICT) Experiment were included in the standard assimilation system at ECMWF.

The PREDICT Experiment, as described in Montgomery et al. (2012), was a dedicated field study that set out to acquire empirical data to quantify thermodynamic and kinematic parameters in developing and non-developing tropical disturbances in the Atlantic Ocean.

10 The primary platform for this experiment was the NSF-NCAR Gulfstream V (GV) with EOL/Vaisala GPS dropsondes. The GV was able to make drops from altitudes as high as $\sim 13$ km. There were 5 research flights with 109 dropsondes conducted during ex-Gaston (Fig. 1).

**4 Results**

15 We begin our analysis by characterizing the vertical shear that affected Gaston's pouch. The vertical shear is calculated in the vicinity of the pouch center, the center being defined here by the intersection of the wave trough and critical latitude at the 700 hPa level. "Deep-layer shear" and "pouch shear" are computed by taking the vector differential of horizontal winds between the 200 and 850 hPa levels, and between the 500 and 850 hPa levels, respectively,

20 averaged over a a $3° \times 3°$ box centered at the pouch center (referred to as pouch-scales). The pouch-scale averaging is performed on a $3° \times 3°$ box, centered on the circulation center as defined by the 700 hPa tracking level.

For both the deep and pouch shear, the magnitude of the shear decreases rapidly from $\sim 20$ m s$^{-1}$ on 30 August to $\sim 2$ m s$^{-1}$ on 2 September (Fig. 2). During the same period, the

25 direction of the deep and pouch shear shifts from easterly to westerly flow (Fig. 3). After 2 September, the magnitude of the shear (deep and pouch) increases to $\sim 5$ m s$^{-1}$. The

pouch shear direction slowly becomes more northerly by 5 September. The deep shear, though, rapidly changes direction from northeasterly to southwesterly from 12:00 UTC 2 September to 00:00 UTC 3 September, in the ECMWF data. The deep shear returns to an easterly flow on 4 September. These shear results are consistent with the analysis of PREDICT data by Davis and Ahijevych (2012). The National Hurricane Center defines vertical shear of 12 m s$^{-1}$ as an upper limit for favorable conditions for tropical cyclogenesis. The magnitude of the vertical shear (typically 4–8 m s$^{-1}$) for ex-Gaston, while below this heuristic limit for a SST of 28.5 °C, does suggest  the potential for a ventilating flow relative to the moving system (Riemer and Montgomery, 2011) and a potential contribution of a dipole-like distribution of vorticity from a non-advective flux (Haynes and McIntyre, 1987; Raymond et al., 2014). This latter contribution could be a net increase or decrease of vorticity.

[revised manuscript text omitted]

Our study of GR14 suggests that these authors appeared to overlook the implications of Davis and Ahijevych (2012) findings of a vertically sheared pouch and RM12's findings of dry air mixing into ex-Gaston's pouch between 1 and 3 September (RM12's Fig. 6), a time period spanning the first day of PREDICT observations (2 September). While GR14 acknowledge the role of a  transient flow component in causing a  in the sign of the low-level vorticity tendency, they  appear to not recognize that this same time-dependence can cause an intrusion of dry air  to enter a pouch that is apparently closed in an instantaneous snapshot. 
[revised manuscript text omitted]

Reasor, P. D., Montgomery, M. T., and Grasso, L. D.: A new look at the problem of tropical cyclones in vertical shear flow: Vortex resiliency, J. Atmos. Sci., 61, 3–22, doi:10.1175/1520-0469(2004)061,0003:ANLATP.2.0.CO;2., 2004.

Riemer, M. and Montgomery, M. T.: Simple kinematic models for the environmental interaction of tropical cyclones in vertical wind shear, Atmos. Chem. Phys., 11, 9395–9414, doi:10.5194/acp-11-9395-2011, 2011.

Rutherford, B., and M. T. Montgomery, 2012: A Lagrangian analysis of a developing and non-developing disturbance observed during the PREDICT experiment. *Atmos. Chem. Phys.*, **12**, 11 355–11 381, doi:10.5194/acp-12-11355-2012.

Samelson, R. M. and Wiggins, S.: Lagrangian Transport in Geophysical Jets and Waves: The Dynamical Systems Approach, vol. 31, Springer Science & Business Media, Berlin, Germany, 2006.

Smith, R. K. and Montgomery, M. T.: Observations of the convective environment in developing and non-developing tropical disturbances, Q. J. Roy. Meteor. Soc., 138, 1721–1739, doi:10.1002/qj.1910, 2012.

Wang, Z., Montgomery, M. T., and Dunkerton, T. J.: Genesis of Pre-Hurricane Felix (2007). Part II: Warm core formation, precipitation evolution, and predictability, J. Atmos. Sci., 67, 1730–1744, doi:10.1175/2010JAS3435.1, 2010a.

Wang, Z., Montgomery, M. T., and Dunkerton, T. J.: Genesis of Pre-Hurricane Felix (2007). Part I: The role of the easterly wave critical layer, J. Atmos. Sci., 67, 1711–1729, doi:10.1175/2009JAS3420.1, 2010b.

Discussion Paper | Discussion Paper | Discussion Paper | Discussion Paper |

**Table 1.** Summary of Dropsondes from PREDICT Research Flight 9 (RF09) on 2 September.

| Drop Num. | Time (UTC) | TPW (kg m$^{-2}$) | CAPE (J kg$^{-1}$) | CIN (J kg$^{-1}$) |
|-----------|------------|-------------------|--------------------|--------------------|
| 1 | 15:32 | 33.0 | 478 | 149 |
| 2 | 15:44 | 48.2 | 196 | 95 |
| 3 | 15:55 | 53.6 | 24 | 142 |
| 4 | 16:05 | 61.1 | 688 | 6 |
| 5 | 16:14 | 62.8 | 706 | 0 |
| 6* | 16:24 | 57.5 | 1047 | 9 |
| 7* | 16:37 | 58.4 | 612 | 29 |
| 8* | 16:47 | 63.0 | 1707 | 0 |
| 9* | 16:54 | 65.9 | 654 | 11 |
| 10* | 17:03 | 67.1 | 1649 | 0 |
| 11* | 17:13 | 59.9 | 1566 | 0 |
| 12* | 17:23 | 57.5 | 605 | 5 |
| 13 | 17:33 | 56.7 | 2 | 158 |
| 14 | 17:45 | 55.3 | 0 | 328 |
| 15 | 17:55 | 53.8 | 114 | 110 |
| 16 | 18:08 | 51.1 | 1155 | 14 |
| 17 | 18:18 | 35.6 | 525 | 75 |
| 18 | 18:30 | 36.1 | 285 | 143 |
| 19 | 18:43 | 38.2 | 101 | 155 |

* denotes dropsonde is in the pouch-scale analysis box. Adapted from Smith and Montgomery (2012).

[revised manuscript text omitted]